# Characterization and Evaluation of Rapamycin-Loaded Nano-Micelle Ophthalmic Solution

**DOI:** 10.3390/jfb14010049

**Published:** 2023-01-16

**Authors:** Ting Zhang, Chao Wei, Xianggen Wu, Sai Zhang, Fangnan Duan, Xiaolin Qi, Weiyun Shi, Hua Gao

**Affiliations:** 1Eye Hospital of Shandong First Medical University (Shandong Eye Hospital), Eye Institute of Shandong First Medical University, Jinan 250021, China; 2State Key Laboratory Cultivation Base, Shandong Provincial Key Laboratory of Ophthalmology, Eye Institute of Shandong First Medical University, Qingdao 266071, China; 3School of Ophthalmology, Shandong First Medical University, Jinan 250021, China; 4College of Chemical Engineering, Qingdao University of Science and Technology, Qingdao 266042, China

**Keywords:** rapamycin, nano-micelle, ocular drug delivery, eye drops, corneal allograft rejection

## Abstract

Rapamycin-loaded nano-micelle ophthalmic solution (RAPA-NM) offers a promising application for preventing corneal allograft rejection; however, RAPA-NM has not yet been fully characterized. This study aimed to evaluate the physicochemical properties, biocompatibility, and underlying mechanism of RAPA-NM in inhibiting corneal allograft rejection. An optimized RAPA-NM was successfully prepared using a polyvinyl caprolactam–polyvinyl acetate–polyethylene glycol (PVCL-PVA-PEG) graft copolymer as the excipient at a PVCL-PVA-PEG/RAPA weight ratio of 18:1. This formulation exhibited high encapsulation efficiency (99.25 ± 0.55%), small micelle size (64.42 ± 1.18 nm), uniform size distribution (polydispersity index = 0.076 ± 0.016), and a zeta potential of 1.67 ± 0.93 mV. The storage stability test showed that RAPA-NM could be stored steadily for 12 weeks. RAPA-NM also displayed satisfactory cytocompatibility and high membrane permeability. Moreover, topical administration of RAPA-NM could effectively prevent corneal allograft rejection. Mechanistically, a transcriptomic analysis revealed that several immune- and inflammation-related Kyoto Encyclopedia of Genes and Genomes (KEGG) pathways were significantly enriched in the downregulated genes in the RAPA-NM-treated allografts compared with the rejected allogenic corneal grafts. Taken together, these findings highlight the potential of RAPA-NM in treating corneal allograft rejection and other ocular inflammatory diseases.

## 1. Introduction

As a classic macrolide immunosuppressant, rapamycin (RAPA) has been widely applied to prevent allograft rejection following transplantation [1,2]. In addition to its anti-rejection effect, numerous therapeutic effects of RAPA have been well documented, including anti-angiogenesis, anti-fungal, anti-tumor, anti-aging, and inflammation resolution [3,4,5,6]. Because of its multi-therapeutic properties, RAPA has received a great deal of attention and been broadly evaluated for treatment of various ocular diseases, such as age-related macular degeneration, uveitis, diabetic macular edema, dry eye, and choroid neovascularization [7,8,9,10,11].

Although the clinical application prospect of RAPA in ocular diseases has drawn enormous attention, several existing drawbacks limit its clinical usage to a large extent. Due to its lipophilic and hydrophobic properties as well as the limitation of the unique anatomic structure of the eye, topical usage of RAPA cannot easily permeate the corneal barrier, and thus an effective drug concentration could not be reached. Similarly, owing to the blood–ocular barrier, systemic administration of RAPA also does not obtain an effective drug concentration in the eye. 

Various techniques and systems have been developed to improve the solubility, stability, and bioavailability of poorly soluble substances, including nanosuspension, micronization, co-solvency, solid lipid nanoparticles, and micellar formulation [12]. As a powerful alternative approach, micellar solubilization has been used to enhance the bioavailability of some products, including curcumin, resveratrol, and paclitaxel [13,14,15]. Polymeric micelles have aroused great concern due to their potential to overcome anatomical barriers, as well as reduce toxicity, and increase ocular retention [16,17,18]. In our previous study, the RAPA-loaded nano-micelle ophthalmic solution (RAPA-NM), fabricated by encapsulation of RAPA into a polyvinyl caprolactam–polyvinyl acetate–polyethylene glycol (PVCL-PVA-PEG) graft copolymer, had an excellent anti-corneal allograft rejection outcome [19,20]. Nonetheless, the physicochemical properties, biocompatibility, and underlying anti-rejection mechanism of RAPA-NM have not yet been sufficiently explored; therefore, we aimed to further evaluate these properties and pave the way for its application. 

## 2. Materials and Methods

### 2.1. Chemical Reagents 

RAPA was purchased from Med Chem Express Co. Ltd. (Monmouth Junction, NJ, USA). PVCL-PVA-PEG was provided by the BASF Corporation (Shanghai, China), and benzalkonium chloride (BAC) was obtained from Sigma-Aldrich Co. Ltd. (St. Louis, MO, USA). The methanol and acetonitrile used were of high-performance liquid chromatography (HPLC) grade, and the other reagents were of analytical grade.

### 2.2. Preparation of RAPA Micelles

RAPA-loaded PVCL-PVA-PEG micelles were prepared using a thin-film hydration method, as previously reported [21]. Briefly, a certain amount of PVCL-PVA-PEG copolymer was added to 50.0 mg of RAPA at different weight ratios (6:1, 9:1, 12:1, 15:1, and 18:1) and co-dissolved in 2 mL of dehydrated ethanol. The solvent was subsequently evaporated under reduced pressure at 40 °C to obtain a thin film on the wall of the flask. Then, the film was hydrated with 10 mL of phosphate-buffered saline (PBS) at 40 °C under moderate shaking, producing a RAPA micelle formulation with different weight ratios of PVCL-PVA-PEG/RAPA. The physicochemical characterization of RAPA micelles was detected to screen the optimal weight ratio of PVCL-PVA-PEG/RAPA.

### 2.3. Properties of the RAPA Micelles

#### 2.3.1. Entrapment Efficiency

The entrapment efficiency (EE) of RAPA in micelles was determined as previously reported [22]. In brief, the RAPA micelles were passed through a 0.22 µm filter to remove the free RAPA. A volume of 100 µL of RAPA micelles was added to 900 µL of methanol, and then the mixtures were vortexed for 1 min. Subsequently, a further 10-fold dilution with methanol was tested by HPLC (Agilent 1200, Santa Clara, CA, USA). HPLC analysis was performed on a reversed-phase C18 column (particle size of 5 μm, 4.6 mm × 150 mm). The mobile phase consisted of acetonitrile–methanol–water (30:35:35 *v*/*v*/*v*) with a flow rate of 1.0 mL/min, while the detection wavelength was set at 278 nm. The sample injected volume and column temperature were 10 μL and 40 °C, respectively. The concentration of RAPA was determined by interpolation in the calibration curve (*Y* = 14.55*x* − 0.24, R^2^ = 0.994). The EE (%) of RAPA was calculated using the following equation: EE% = (RAPA_total_ − RAPA_free_)/RAPA_total_, in which RAPA_total_ and RAPA_free_ represent the initial and free RAPA concentrations, respectively. 

#### 2.3.2. Micelle Size and Zeta Potential

Dynamic light scattering (DLS) is a non-destructive technique to measure molecules at the sub-micron level. The particle size, polydispersity index (PDI), and zeta potential of RAPA micelles were examined by DLS using a Zetasizer Nano ZS90 analyzer (Malvern Instruments Ltd., Malvern, UK). The DLS measurements were performed with a scattering angle of 90° using a laser wavelength of 659.0 nm at the temperature of 25 °C. The micelles used were diluted at a concentration of 1 mg/mL of RAPA. 

### 2.4. Physicochemical Characterization of the Optimal RAPA Micelles

#### 2.4.1. Transmission Electron Microscopy (TEM)

The micelles consisting of PVCL-PVA-PEG/RAPA at an 18:1 weight ratio displayed excellent characteristics and were used for subsequent studies. The RAPA micelles were morphologically characterized using TEM (JEM-1200EX, Jeo, Tokyo, Japan). For the TEM examination, the samples were stained with phosphotungstic acid solution for 2 min. Then, a drop of the specimen was placed in a carbon-coated copper mesh, and the excess solution was blotted by a filter paper. After being air-dried thoroughly, the samples were photographed under 200,000× magnification (at a voltage of 10 KV).

#### 2.4.2. Differential Scanning Calorimetry (DSC)

DSC were performed through the free RAPA powder, PVCL-PVA-PEG, physical mixture of RAPA and PVCL-PVA-PEG, and lyophilized RAPA micelles as previously described [23]. Samples weighing 5–10 mg were added to the aluminum pans and heated from 25 to 300 °C at a rate of 10 °C/min in a nitrogen atmosphere (flow rate 100 mL/min). DSC data were measured by a DSC204F1 differential scanning calorimeter (NETZSCH Group, Selb, Germany). 

#### 2.4.3. Infrared (IR) Spectrophotometry

To determine whether the chemical and physical interactions present between the RAPA and polymer, the IR spectra were collected by a VERTEX70 IR spectrophotometer (Bruker Corporation, Bremen, Germany) using the free RAPA reagent, PVCL-PVA-PEG, physical mixture of RAPA and PVCL-PVA-PEG, and lyophilized RAPA micelles as previously described [24]. The wavenumber range was set from 4000 to 400 cm^−1^. The number of scans was 16 times, and the resolution was 4 cm^−1^.

#### 2.4.4. X-Ray Diffraction (XRD)

To confirm if crystallographic RAPA exists in the micelles, XRD analysis was conducted by a Bruker D8 ADVANCE diffractometer (Bruker Corporation, Bremen, Germany) using the free RAPA powder, PVCL-PVA-PEG, physical mixture of RAPA and PVCL-PVA-PEG, and lyophilized RAPA micelles as previously described [23,24]. Briefly, the measurements were performed using Cu Ka radiation with the scanned angle from 5° to 60°.

### 2.5. Storage Stability Test

RAPA-NM were prepared and packaged in sterile glass vials for storage stability evaluation. In brief, the samples were kept from light for up to 12 weeks at 25 °C and 4 °C. The particle size, PDI, zeta potential, and amount of RAPA remaining in micelles were determined every 2 weeks [25]. 

### 2.6. Cytocompatibility Evaluation

The cytocompatibility of RAPA micelles to human corneal epithelial cells (HCECs) was evaluated using the methyl thiazolyl tetrazolium (MTT) assay, as previously described [23]. Immortalized HCECs were gifted by professor Chonn-Ki Joo of the Catholic University of Korea, Seoul, Korea. H. Cells were seeded in a 96-well plate, and cultured in DMEM/F12 (Gibco, Grand Island, NY, USA) with 10% FBS at 37 °C in a humidified atmosphere containing 5% carbon dioxide (CO_2_). For the long-term cytocompatibility evaluation, the medium was replaced with 200 µL of fresh culture medium containing different concentrations of RAPA-NM (1.56 μg/mL, 3.13 μg/mL, 6.25 μg/mL, 12.5 μg/mL, 25 μg/mL, 50 μg/mL, 100 μg/mL, 250 μg/mL, and 500 μg/mL) for 24, 48, and 72 h. For the short-term observation, cytocompatibility of the RAPA-NM (2.5 mg/mL and 5 mg/mL) was tested following 1 h incubation, with BAC (50 μg/mL and 100 μg/mL) serving as the positive control. At scheduled time points, the medium was removed and cells were washed with PBS and then incubated with 200 µL of fresh medium containing 20 µL of MTT solution. Next, the plates were incubated for an additional 4 h at 37 °C. After the medium with MTT was removed, 150 µL of dimethyl sulfoxide (DMSO) was added to each well. Thereafter, the plates were incubated for 10 min to dissolve the formazan crystals, and the optical density was measured at a wavelength of 490 nm using the Multiskan GO instrument (Thermo Scientific, Waltham, MA, USA).

### 2.7. In Vitro Parallel Artificial Membrane Permeability Assay

The parallel artificial membrane permeability assay (PAMPA), which enables fast evaluation of the ability of formulations to permeate membranes by passive diffusion, is applied to determine membrane and transcellular permeation of potential drugs. This membrane permeation has also indicated the potential of improved in vivo corneal permeation [26,27]. The PAMPA experiment was performed as previously reported [28,29,30]. Briefly, free RAPA was dissolved in DMSO and further diluted with an artificial tears solution to obtain a RAPA concentration of 5 mg/mL. Then, RAPA micelles were diluted with an artificial tear solution containing DMSO to form a RAPA concentration of 5 mg/mL. The final DMSO concentration of these two samples was 5% (*v*/*v*). Solutions were added to donor wells at a volume of 0.5 mL. The acceptor wells were filled with 0.5 mL of blank artificial tear solution. The PAMPA plates were incubated at 25 °C for 3.5 h. Solutions from the acceptor wells were collected and transferred into a volumetric tube to obtain a constant volume before analysis via HPLC.

### 2.8. Animal Experiments

All animal experiments were approved by the Ethics Committee of the Shandong Eye Institute (Approval Code: G-2015-005), and all procedures were performed according to the guidelines of the Association for Research in Vision and Ophthalmology. Six- to eight-week-old male BALB/C and C57BL/6 mice were obtained from Charles River (Beijing, China) and used to establish an allogenic corneal transplantation model. All procedures were performed on one eye of each animal under systemic and topical anesthesia, with all efforts to minimize suffering. The mice were sacrificed by anesthetic overdose at the end of the experiment. 

### 2.9. In Vivo Anti-Inflammatory Activity Evaluation

The RAPA micelles were filtered through a 0.22 µm filter for sterilization and were then further diluted with PBS to prepare a 0.1% RAPA nano-micelle ophthalmic solution (1 mg/mL). To explore the anti-inflammatory activity of the RAPA-NM, we established a murine orthotopic penetrating keratoplasty model. In brief, the mice were randomly divided into three groups: a normal group that was not subjected to any procedure (Nor, *n* = 12), an allogenic corneal transplantation group (Allo, *n* = 12), and an allogenic corneal transplantation group receiving RAPA-NM (1 mg/mL) four times every day (RAPA, *n* = 12). Keratoplasty was performed as previously described [31]. In brief, the donor corneas of the C57BL/6 mice were incised by a 2.25 mm trephine to produce a corneal button graft. The BALB/C recipient corneas were excised centrally after being cut with a 2 mm trephine. Then, the donor cornea was sutured onto the recipient graft bed with eight interrupted 11-0 nylon sutures (Mani, Inc., Tokyo, Japan). A 0.3% ofloxacin eye ointment (Suzhou, China) was applied at the end of the surgery. Sutures were removed 7 days postoperatively. Eyes with complications of hyphemia, infection, or cataract were excluded from this study. Cornea status was graded for graft survival using slit lamp microscopy according to a previous criterion [32].

### 2.10. Transcriptome Sequencing

To elucidate the effects of RAPA-NM on corneal allograft rejection, we performed transcriptome sequencing using 3-week postoperative corneal tissues. The RNA sequence analysis was performed by Shanghai Kuangchen Biotechnology Co., Ltd., as previously described [33]. In brief, three cornea samples from each group were homogenized with TRIzol reagent (Invitrogen, Carlsbad, CA, USA) to extract the total RNA. An RNA sequencing (RNA-Seq) library was constructed using a KAPA Stranded RNA-Seq Library Prep Kit (Illumina) by NanoDrop ND-1000. The RNA was enriched with an NEB Next^®^ Poly(A) mRNA Magnetic Isolation Module and fragmented into 200 bp to 300 bp pieces by a fragmentation buffer. Next, the RNA was used as a template and reversed into cDNA. After being purified, randomly primed first-strand and dUTP-based second-strand cDNA molecules were synthesized; then, the adaptor was ligated, and the cDNA library was amplified. The total library was quantified by an Agilent 2100 Bioanalyzer and sequenced with Illumina NovaSeq 6000. The raw sequencing data were filtered by Cutadapt and differentially expressed genes (DEGs) with a value of *p* < 0.05 and |log2 fold change| > 1.5 were screened out for the KEGG enrichment analysis. Gene pathways were considered enriched if they had a *p* value < 0.05. KEGG analysis was performed using the clusterProfiler package in R.

### 2.11. Statistical Analysis

All experiments were conducted in triplicate, and the data are presented as mean ± SD. The results were analyzed using Graph Prism 8.0 (GraphPad Software, Inc., San Diego, CA, USA). The graft survival was analyzed by the Kaplan–Meier curve analysis. A two-tailed Fisher’s exact test was employed to test the functional enrichment of the DEGs. In all experiments, *p* < 0.05 was considered statistically significant. 

## 3. Results

### 3.1. Preparation and Characterization of RAPA Micelles

To obtain optimized RAPA-NM, RAPA micelles with different weight ratios of PVCL-PVA-PEG/RAPA were prepared. The results demonstrated that the characterization of RAPA micelles, including micelle size, EE, PDI, and zeta potential, highly depended on the weight ratio of PVCL-PVA-PEG/RAPA (Appendix A). When the weight ratio of PVCL-PVA-PEG/RAPA was 6:1, only 27.45% of the RAPA could be encapsulated into the micelles, but the EE sharply reached 97.98% when the weight ratio of PVCL-PVA-PEG/RAPA was 15:1. However, when the PVCL-PVA-PEG/RAPA weight ratio rate was 18:1, the EE did not significantly increase, reaching a peak of 98.92%. As the weight ratio of PVCL-PVA-PEG/RAPA increased from 6:1 to 15:1, both the micelle size and PDI were decreased; however, these two parameters slightly increased at a weight ratio of 18:1. The zeta potential remained constant regardless of the change in the PVCL-PVA-PEG/RAPA weight ratio. The intra-day and inter-day variations were both less than 5.0%. Based on these findings, a PVCL-PVA-PEG/RAPA weight ratio of 18:1 was identified as the optimal formation for RAPA micelles, with a high EE of 98.92%. Thus, it was used for the subsequent experiment.

### 3.2. Outline of RAPA-NM with PVCL-PVA-PEG/RAPA Weight Ratio of 18:1

We subsequently characterized the profiles of RAPA-NM prepared from an optimized PVCL-PVA-PEG/RAPA weight ratio of 18:1. As shown in Figure 1A, RAPA-NM had a transparent and light white appearance. The TEM image shows that RAPA-NM had good dispersibility in an aqueous solution, and its particles had a spherical shape and smooth surface (Figure 1B). The TEM image further confirmed the results acquired from the DLS, which showed the RAPA-NM had a micelle size of 64.42 ± 1.18 nm and a particle distribution of 0.076 ± 0.016 (Figure 1C). Moreover, RAPA-NM featured a mean zeta potential of −1.67 ± 0.93 mV (Figure 1D). This formulation was selected as the ophthalmic solution for the subsequent tests.

### 3.3. Physicochemical Properties of the RAPA Micelles

DSC, IR, and XRD studies of the free RAPA reagent, PVCL-PVA-PEG, physical mixtures of RAPA and PVCL-PVA-PEG, and lyophilized RAPA micelles were performed. The RAPA thermogram clearly shows an endothermic peak at 195.6 °C (Figure 2A). RAPA’s endothermic peak has been previously reported to be between 183 °C and 205 °C, which corresponds to its melting point [34]. The DSC curve showed that PVCL-PVA-PEG had a relatively mild endothermic peak at 283.3 °C, which was similar to its melting point. For the physical mixture of PVCL-PVA-PEG and RAPA, the melting points of RAPA and PVCL-PVA-PEG remained nearly unchanged. When RAPA was encapsulated into PVCL-PVA-PEG, the endothermic peak of RAPA was absent, whereas the melting range of PVCL-PVA-PEG was extended, indicating the complete embedment of RAPA into the hydrophobic core of the PVCL-PVA-PEG nanoparticles [35]. 

The IR spectra are shown in Figure 2B. The RAPA spectra consisted of the characteristic absorption peaks at 3588.22  cm^−1^ and 3415.05 cm^−1^ (overlapping O–H stretching vibration and N–H stretching vibration), 2931.58−2822.65 cm^−1^ (stretching vibration of methyl and methylene C–H), 1719.82 cm^−1^ and 1635.13 cm^−1^ (C–C stretching vibration of benzene ring skeleton), 1399.90 cm^−1^ (methyl C–H symmetrical bending vibration), 1243.82 cm^−1^ and 1103.85 cm^−1^ (C–O stretching vibration), and 910.91 cm^−1^ (C–H out of plane bending vibration on benzene ring). For PVCL-PVA-PEG, absorption bands appeared at 3447.67 cm^−1^ (O–H stretching vibration of phenols), 2927.59 cm^−1^ (C–H stretching vibration), 1636.61 cm^−1^ (stretching vibration of C=O), and 1481.38 cm^−1^ (stretching vibration of C–O–C). When compared with the IR spectra of RAPA, PVCL-PVA-PEG, and their mixture, the IR spectra of the RAPA micelles showed no new absorption peaks. These results indicated that no chemical reactions occurred during the preparation of RAPA-NM.

The XRD analyses are shown in Figure 2C. The diffraction diagram of RAPA showed multiple and intensive peaks at 7.14°, 10.20°, 14.38°, 16.20°, 19.98°, 20.44°, and so on, indicating that RAPA was preserved in a highly crystalline state. The diffraction image of PVCL-PVA-PEG showed an amorphous state. The physical mixture of PVCL-PVA-PEG and RAPA exhibited the typical bands of PVCL-PVA-PEG, and the RAPA peaks were covered by the excipient. Meanwhile, the characteristic RAPA peaks in the RAPA micelles disappeared. These findings not only demonstrate the existence of RAPA with an amorphous state in RAPA micelles [36], but also reveal the excellent diffusion of RAPA in the PVCL-PVA-PEG, which could manage a sustained release of the encapsulated drug.

### 3.4. Storage Stability

To evaluate the storage stability of RAPA-NM, we stored it at either 4 °C or 25 °C for 12 weeks. We then determined the RAPA dosage, nano-micelle size, PDI, and zeta potential of the solution. The initial RAPA quantities in the ophthalmic solution were set as 100.0%. After 12 weeks of storage, the proportions of RAPA remaining in the ophthalmic solutions were 93.98 ± 4.28% and 90.68 ± 3.91% at 4 °C and 25 °C, respectively (Figure 3A). Moreover, no significant dynamic alterations in particle size, PDI, or zeta potential were observed throughout the 12-week observation period at either 4 °C or 25 °C (Figure 3B–D). Based on these results, 4 °C was considered an optimal temperature for RAPA-NM storage. 

### 3.5. Cytocompatibility Evaluation

To determine the cytocompatibility of RAPA-NM, MTT assay was performed in RAPA-NM-stimulated HCECs. As shown in Figure 4A–C, during the long-term observation, the viability of HCECs decreased with the increased RAPA-NM concentration and prolonged RAPA-NM exposure. A high survival rate (>90%) was observed following incubation with ≤100 μg/mL of RAPA-NM for 24 h and 48 h as well as following incubation with ≤25 μg/mL of RAPA-NM for 72 h. However, when incubated with 500 μg/mL of RAPA-NM for 24 h, 250 μg/mL of RAPA-NM for 48 h, or 50 μg/mL of RAPA-NM for 72 h, the treated HCECs showed lower cellular viability. In the short-term exposure, BAC, which is commonly used in commercial eye drops, served as the positive control. When treated with a BAC of 50 μg/mL and 100 μg/mL for 1 h, the cell viability decreased to 15.43% and 14.47%, respectively. However, when treated with 2.5 mg/mL and 5 mg/mL of RAPA-NM for 1 h, satisfactory cytocompatibility (survival rate > 90%) was observed in HCECs (Figure 4D). 

### 3.6. In Vitro PAMPA 

The PAMPA experiment is widely used as an in vitro model for evaluating membrane permeability and transcellular permeation [26,27,28,29,30]. As shown in Figure 5, compared with the free RAPA solution, the RAPA-NM exhibited a strong permeability potential. After 3.5 h of treatment, the levels of permeated RAPA in the RAPA-NM group (23.84 ± 0.83 μg) were considerably higher than those in the free RAPA solution group (0.45 ± 0.01 μg). The membrane permeability of the RAPA-NM was over 50-fold higher than that of the free RAPA solution. 

The drug release data were applied to various release kinetic models. As shown in Table 1, results indicated that the RAPA-NM fit the zero-order kinetic model (R^2^ = 0.9966), suggesting a constant in vitro permeation mechanism of RAPA micelles. Free RAPA solution followed the first-order kinetic model (R^2^ = 0.8365), suggesting a concentration-dependent in vitro permeation mechanism involved in free RAPA.

### 3.7. Systemically Understanding the Mechanism of RAPA-NM in Inhibiting Corneal Allograft Rejection

A corneal transplantation model was established to investigate the anti-rejection effect of RAPA-NM and to elucidate the underlying mechanism through transcriptomic analysis. Compared with the untreated corneal allografts (Allo group), the RAPA-NM-treated allografts (RAPA group) showed significantly reduced allograft rejection, longer survival time, and nearly transparent corneas (Figure 6). Based on these findings, corneal grafts were harvested from different groups for transcriptomic analysis and subsequent mechanism investigation. The principal component analysis (PCA) revealed that the samples obtained from the Nor, Allo, and RAPA groups could be separated into three clusters based on their gene expression profiles (Appendix A). Compared with the Nor group, the Allo group had a total of 3383 DEGs, of which 1951 were upregulated and 1432 were downregulated (Appendix A). Compared with the Allo group, the RAPA group had a total of 737 upregulated and 1058 downregulated genes. The clustering heatmap analysis revealed the significant discrepancy among these groups in terms of their gene-expressing profiles (Figure 7A). 

To determine the mechanisms of anti-rejection by RAPA-NM, we performed the KEGG pathways analysis. Compared with the normal corneas, the upregulated genes in rejected corneal grafts were functionally associated with numerous inflammation- and immune-related pathways, including cytokine–cytokine receptor interaction, chemokine signaling, antigen processing and presentation, graft-versus-host disease, phagosome, NF-kappa B signaling, allograft rejection, and Th1 and Th2 cell differentiation. Compared with the rejected allografts, numbers of inflammation- and immune-related KEGG pathways in the RAPA-NM-treated corneal grafts were enriched in the downregulated genes, which were mainly involved in cytokine–cytokine receptor interaction, chemokine signaling, T-cell receptor signaling pathways, graft-versus-host disease, phagosome, NF-kappa B signaling, and Th1 and Th2 cell differentiation (Figure 7B). These results indicated that the boosted inflammation and alloimmune activity in the rejected corneal allografts were pronouncedly blockaded by RAPA-NM, which likely contributed to the delayed corneal allograft rejection.

## 4. Discussion

The strong immunosuppressive effect of RAPA renders this drug a promising agent for treating ocular diseases, including corneal allograft rejection [37,38,39]. However, the lipophilic and hydrophobic properties of RAPA and the existence of the blood–ocular barrier and corneal barrier greatly limit its clinical application in ophthalmology [40]. Encapsulation of the hydrophobic drugs into nanoparticles has proven to be an effective strategy for improvement of their solubility, stability, and bioavailability [41,42]. Here, we successfully developed an ophthalmic delivery system for RAPA using PVCL-PVA-PEG, with high encapsulation efficiency, good solubility and stability, and satisfactory bioavailability. RAPA-NM also significantly alleviated corneal allograft rejection by inhibiting inflammation- and immune-related signaling. 

PVCL-PVA-PEG is an FDA-approved novel block copolymer which is soluble in both water and organic solvents and has a molecular weight of 90 to 140 kDa [43]. It has been applied to increase the solubility and bioavailability of poorly soluble molecules through micelle formation [44,45,46]. We successfully developed RAPA-NM and demonstrated its good solubility, stability, and bioavailability. These findings are supported by a previous study, which reported that PVCL-PVA-PEG micelles loaded myricetin with excellent solubility and stability, improved permeation, and showed no irritation or toxicity [20]. Moreover, several RAPA nano-formulations have been developed using polylactic acid (PLA), poly lactic-co-glycolic acid (PLGA), and Eudragit RS with particle sizes ranging from 180 nm to 280 nm [47,48]. Comparatively, RAPA-NM at a PVCL-PVA-PEG/RAPA weight ratio of 18:1 had a mean particle size of 64.42 nm. Cumulative evidence reveals that a smaller particle size means higher penetration capacity [49]. In this regard, RAPA-NM demonstrates a great penetration potential, highlighting its promising clinical applications. 

Long-term stable storage is indispensable for drug delivery systems. RAPA nano-formulations using N-palmitoyl-N-monomethyl-N, N-dimethyl-N, N,N-trimethyl-6-O-glycolchitosan (GCPQ) as carriers have been reported. The GCPQ-loaded RAPA remained stable for one month at 5 °C and room temperature [50]. Noteworthily, RAPA remaining in the RAPA-NM was as high as 93.98 ± 4.28% after 12 weeks of storage at 4 °C, with no significant dynamic alterations in particle size, PDI, or zeta potential, demonstrating an ideal storage stability. The superior storage stability of the RAPA-NM, as well as its high encapsulation efficiency, is likely attributed to the low critical micelle concentration of PVCL-PVA-PEG. Moreover, it is worth noting that the PVCL-PVA-PEG chains can orient themselves to give the molecule a structure suitable for the formulation of solid dispersions and solutions [51]. The compatible structure of PVCL-PVA-PEG likely provided RAPA-NM with a high encapsulation efficiency and long-term stability.

An excellent ophthalmic solution should easily penetrate the ocular barrier and deliver the drug to target region without tissue damage. The PAMPA test showed that the membrane permeability of RAPA-NM was more than 50-fold higher than that of the free RAPA solution, suggesting that RAPA-NM with superior transcellular permeation allowed RAPA to penetrate the corneal barrier more easily. Moreover, the MTT test illustrated that HCECs had good cellular tolerance to RAPA-NM regarding both short- and long-term exposure. Besides, our previous findings proved the non-irritation of RAPA-NM on the ocular surface using a modified Draize test [19]. Together, these findings reveal that RAPA-NM holds satisfactory permeability and biocompatibility as an ophthalmic solution. However, future studies are required to determine its permeability in vivo, as well as the distribution and concentration of RAPA in different ocular tissues. 

Graft rejection is the main reason for corneal transplantation failures, and the underlying mechanism remains largely unknown [52]. Our previous findings depicted the advantageous anti-rejection effect of RAPA-NM both in the murine corneal transplantation model and rabbit high-risk corneal transplantation model [19,20], which underscored its promising application prospects. Nevertheless, its anti-rejection mechanism has not yet been systemically investigated. Through transcriptomic analysis, we found numerous immune- and inflammation-related KEGG pathways were significantly enriched in the downregulated genes of RAPA-NM-treated corneal allografts, such as the IL-17 signaling pathway and the NOD-like receptor signaling pathway. Several studies have proved that elevated NLRP3 inflammasome (a NOD-like receptor) activity exacerbated corneal allograft rejection, while autophagy enhanced by RAPA-NM significantly prolonged the survival of corneal allografts by blocking the NLRP3 inflammasome [53,54]. These results support the findings acquired by transcriptomic analysis in this study. 

In conclusion, the RAPA-NM developed by PVCL-PVA-PEG exhibited high RAPA encapsulation efficiency, good solubility, and long-term storage stability. RAPA-NM also displayed excellent cytocompatibility and enhanced transcellular permeation potential. As expected, the topical administration of RAPA-NM achieved significant anti-corneal allograft rejection outcomes mechanistically associated with the blockade of several inflammation- and alloimmune-related KEGG pathways. These findings demonstrate that RAPA-NM with satisfactory solubility and biocompatibility holds promising potential for treating corneal allograft rejection and other ocular inflammatory/autoimmune diseases. 

## Figures and Tables

**Figure 1 jfb-14-00049-f001:**
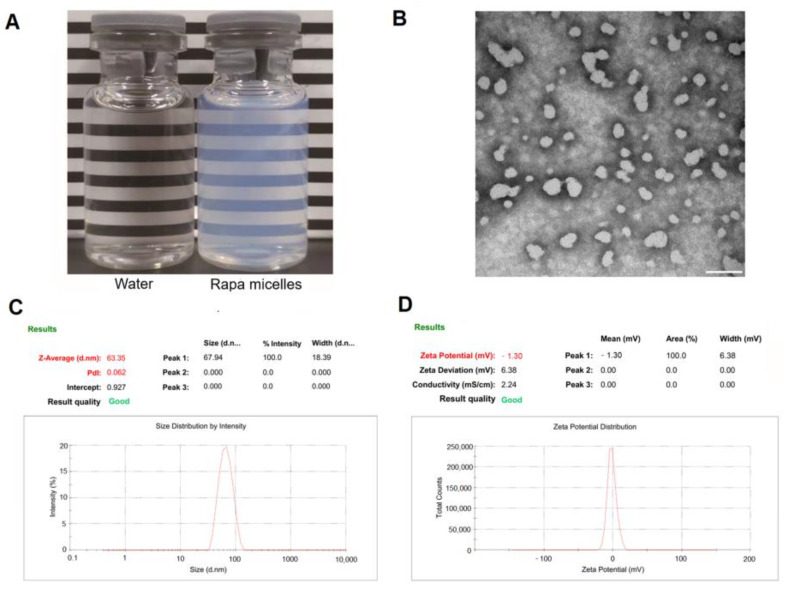
Outline of RAPA-NM with a PVCL-PVA-PEG/RAPA weight ratio of 18:1. (**A**) Appearance, (**B**) morphology as characterized by TEM, (**C**) particle size, and (**D**) zeta potential as determined by DLS. 200,000× magnification. Scale bar = 100 nm.

**Figure 2 jfb-14-00049-f002:**
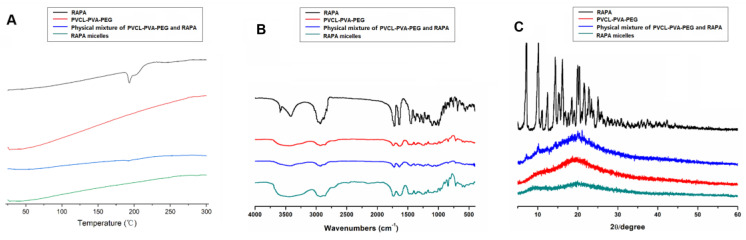
Physicochemical properties of the RAPA micelles. Physicochemical properties as characterized by (**A**) DSC, (**B**) IR, and (**C**) XRD analyses performed for RAPA, PVCL-PVA-PEG, the physical mixture of PVCL-PVA-PEG and RAPA, and RAPA micelles.

**Figure 3 jfb-14-00049-f003:**
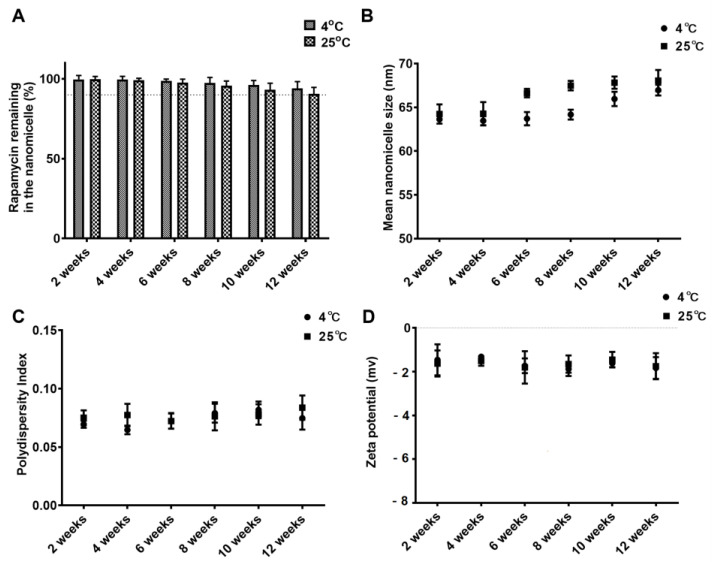
Characterization of RAPA-NM storage under different conditions. The (**A**) quantity, (**B**) particle size, (**C**) PDI, and (**D**) zeta potential of RAPA-NM measured every 2 weeks for 12 weeks at 4 °C or 25 °C (*n* = 6).

**Figure 4 jfb-14-00049-f004:**
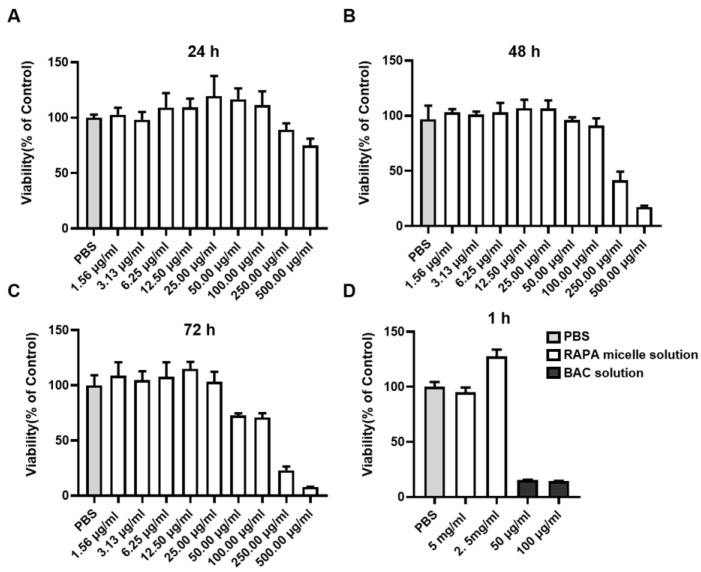
Cytocompatibility examination using MTT assay. Viability of HCECs stimulated for (**A**) 24 h, (**B**) 48 h, and (**C**) 72 h with different concentrations of RAPA-NM; (**D**) the survival rate of HCECs incubated with different concentrations of RAPA-NM or BAC for 1 h (*n* = 4).

**Figure 5 jfb-14-00049-f005:**
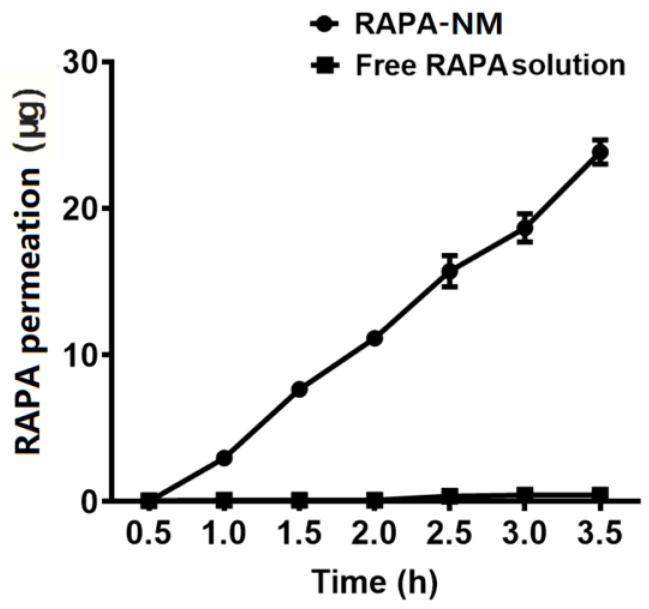
Transcellular permeation of RAPA-NM using PAMPA. Transcellular permeation of the RAPA-NM and free RAPA solution was evaluated by PAMPA using a transwell method (*n* = 6).

**Figure 6 jfb-14-00049-f006:**
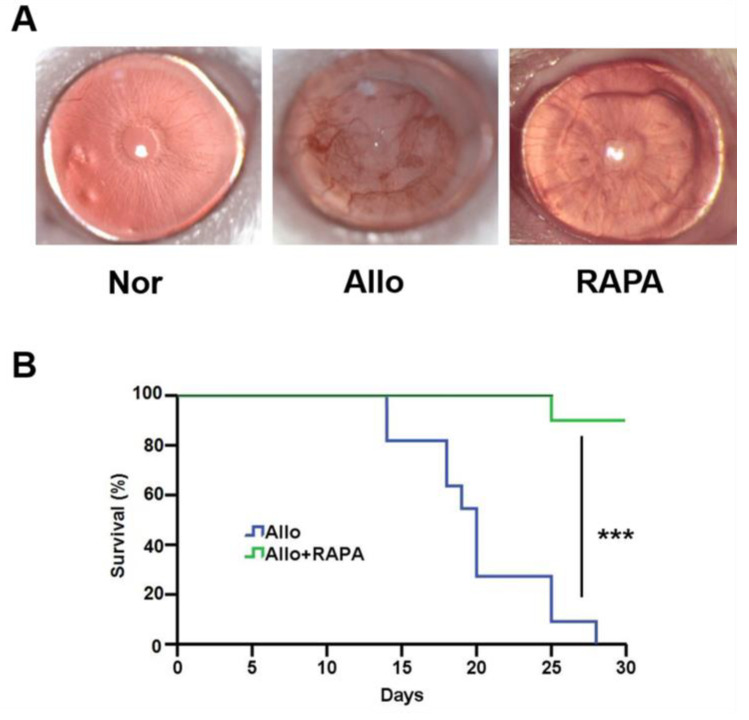
Anti-rejection effect of the RAPA-NM. (**A**) Representative slit lamp photographs of the cornea obtained from different groups. (**B**) Corneal allograft survival curve, *n* = 12, *** *p* < 0.001.

**Figure 7 jfb-14-00049-f007:**
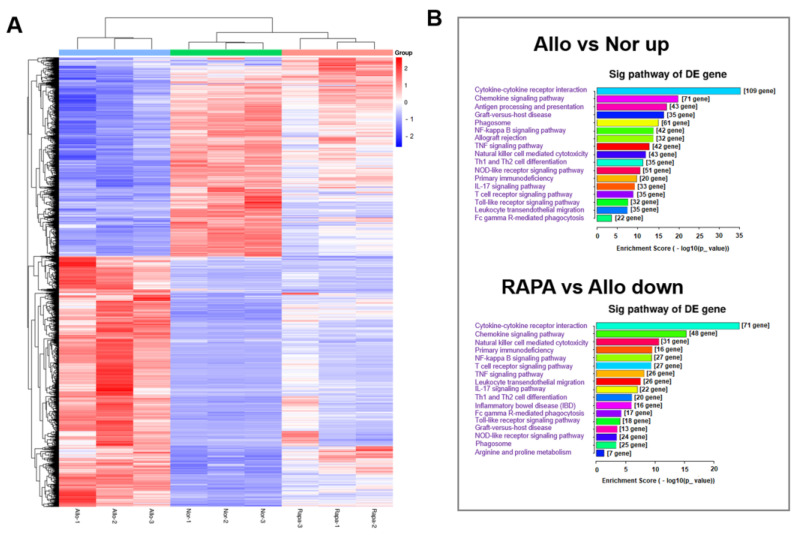
Clustering heatmap of DEGs and analysis of inflammation- and immune-related Kyoto Encyclopedia of Genes and Genomes (KEGG) pathways. (**A**) Heatmap showing the DEGs in the corneas of different groups. (**B**) The enriched KEGG pathways in the upregulated genes of rejected corneal grafts vs. the enriched KEGG pathways in the downregulated genes of RAPA-NM-treated corneal grafts.

**Table 1 jfb-14-00049-t001:** Results of fitting models.

	RAPA-NM		Free RAPA Solution	
	Fitted Equation	R^2^	Fitted Equation	R^2^
Zero order	*Q* = 7.9206*t* − 4.4081	0.9966	*Q* = 0.1498*t* − 0.0649	0.8363
First order	Ln(100 − *Q*) = − 0.09*t* + 4.6598	0.993	Ln(100 − *Q*) = − 0.0015*t* + 4.6058	0.8365
Higuchi	*Q* = 20.446*t*^1/2^ − 16.403	0.8992	*Q* = 0.3763*t*^1/2^ − 0.2776	0.7730
Korsmeyer–Peppas	Lg*Q* = 3.1718Lg*t* − 0.0474	0.9622	Lg*Q* = 1.0583Lg*t* − 0.9925	0.7684
Hixson–Crowell	(100 − *Q*)^1/3^ = − 0.1334*t* + 4.7203	0.7770	(100 − *Q*)^1/3^ = − 0.0023*t* + 4.6426	0.8365

*Q*: RAPA cumulative release (%); *t*: time.

## Data Availability

The data that support the findings of this study are available from the corresponding author upon reasonable request.

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
