# Peer review of "Characterization and Evaluation of Rapamycin-Loaded Nano-Micelle Ophthalmic Solution"

_jfb, 2023, doi:10.3390/jfb14010049_

Round 1

Reviewer 1 Report

Dear Authors,

while not bringing any particular novelties, the paper is interesting.

Reviewer 2 Report

The work has been improved on the basis of the referees' suggestions but there are some parts that have not been expanded and justified as necessary. As regards PAMPA, a justification of the use of this artificial membrane as a model of corneal epithelium was requested, since on the basis of  its composition, it is suitable for simulating other tissues but not the ocular one. The literature that the authors reported does not pertain to ocular use. Furthermore, the authors still do not report a thorough discussion of the results.

Reviewer 3 Report

The manuscrip titled “Characterization and evaluation of rapamycin-loaded nano-micelle ophthalmic solution” by Ting Zhang et al. was already submitted to Pharmaceutics in July 2022 as authors state in their cover letter. However, despite they also declare that the manuscript has been improved according to previous reviewers’ comments and suggestion it is not always true.

Below I report the previous queries which were not addressed, the responses and my actual comments:

ABSTRACT:

Reviewer previous queries:

-          The abstract is too long and it does not meet the guideline for authors (max 200 words).

-          Line 16 and also below: do you mean physiochemical or physico-chemical properties?

-          Line 22: what do you mean with “stable zeta potential”?

Response: Thanks. We have revised the abstract part, The content mentioned has been modified to

physiochemical and zeta potential respectively in line 19 and line 24.

Reviewer actual queries:

-          The mentioned revision of the abstract part consisted in changing just two words. However, the abstract still remained too long and it does not meet the guideline for authors (max 200 words), I still do not understand the authors concept of physiochemical or physico-chemical properties.

INTRODUCTION:

Reviewer previous queries:

-          It should be rephrased as the quality and the structure of the sentences is quite low and sometimes it thus become low understandable.

-          Punctuation check required

Response: Thank you for your kind advice. We emended the introduction part of the manuscript.

Reviewer actual queries:

-          Despite author statement the introduction was not modified at all.

MATERIALS AND METHODS:

Reviewer previous queries:

-          The number of repetition of each experiment is missed.

-          Line 96: the details about the employed HPLC instrument, the chromatographic parameters and the calibration curve are missed.

-          Paragraph 2.5 should be divided into different paragraphs for each characterization as these are the key point of the paper.

-          Line 105: details about the DLS parameters are missed.

-          Details about the XRD, DSC and IR analysis are missed.

-          Line 124: the formulation is not a solution. It is a colloidal dispersion. Correct all along the manuscript.

-          In paragraph 2.7 details are missed.

-          Due to results it should be better to talk about cytocompatibility instead of cytotoxicity.

-          Paragraph 2.8: a short description of the experimental set up is required.

-          The number of animals involved in the study and the animal groups are missed.

Responses and actual queries:

-          All experiments were performed on at least three occasions.

This information must be added in the experimental section. Usually it should be stated “experiments were performed in triplicate and results are expressed as means ± SE or SD”.

-          We added relative description in line 83-89. The details of HPLC can be referred to our established method [13], and concentration of RAPA in the solution was determined by interpolation in the average calibration curve, with dilution factor was considered for the calculation.

Details are still missed: model, manufacturer and characteristics of the employed HPLC as well as information regarding the calibration curve (λ, linearity range, regression analysis curve, R and SE), information about any eventual interference and regarding interday and intraday variations.

-          To acquire a better clarify, we divided it into two parts: Properties of the RAPA micelles, and Physicochemical characterization of the optimal RAPA micelles, which were showed in line 81-109.

It is still confusing. As the characterization are a central part of the work each characterization should deserve a own paragraph. Thus the following 6 paragraphs are expected: EE%; DLS and Z potential; TEM; DSC; IR; XRD.

-          Details about the DLS were descried in line 90-92.

Details are still missed: e.g. scattering angle, λ of the laser source, temperature.

-          The DSC tests have been described further in line 102-106. The procedures of IR and XRD is similar to the referred literature [18].

The DSC improvements are ok. However the IR and XRD parts need to be extensively improved.

-          Thank you, we have corrected the use of formulation in the whole manuscript.

This is the main crucial and serious concept problem. As I already said the formulation is a micellar colloidal dispersion which is not a solution!!!

-          The cytotoxicity of RAPA micelles to human corneal epithelial cells was described and conducted as previously described [17]. It has been rephrased as the quality and the structure of the sentences is low understandable in line 117-120.

Even though I suggested to talk about cytocompatibility instead of cytotoxicity authors still talk about cytotoxicity. Additionally some details are still missed.

-          The paragraph 2.8 was parallel artificial membrane permeability test in the last version of manuscript, we have added the details in line 125-131.

Some details are still missed: e.g. withdrawn volume, timing etc.

-          The number of animals involved and the animal groups are supplemented in line 144-146.

These information are still missed.

RESULTS AND DISCUSSION:

-          A carefully check is still needed.

ALL ALONG THE MANUSCRIPT:

-          Language and style must still be enhanced.

Reviewer 4 Report

The authors have significantly improved the manuscript, only a minor comment is noted in the introduction. The last paragraph (page 2) begins to describe the results and therefore should be cut at line 59.

Round 2

Reviewer 2 Report

accepted

Reviewer 3 Report

Authors have considerably tryied to improve their paper and they have addressed the majority of my previous queries. However, some points still need to be considered:

- the introduction is not informative enough. More information regarding the literature in terms of recently published works proposing polymeric micelles for ocular application should be added (e.g., 10.1016/j.jconrel.2021.06.011, 10.1016/j.msec.2021.111890).

- HPLC condition: the ratio of the employed solvents is still missed.

- paragraphs 2.3 and 2.4 should be divided into more sections in order to better put in evidence the number of characterizations conducted on the proposed micelles.

- IR analysis: the number of scans is still missed.

- authors still refer to "micelle solution" which is a serious mistake! About 60 nm micelles do not form a solution but a colloidal dispersion

- details about the cytocompatibility assay are still missed. As an example 10.1016/j.xphs.2018.09.034

- paragraph 2.7: how are you sure that the dilution of lyophilized micelles with DMSO does not damage the resulting micellar dispersion??

- permeability through PAMPA could be matematically elaborated thus resulting in Js and Kp comparison
